# Anticancer Therapeutic Strategies Targeting p53 Aggregation

**DOI:** 10.3390/ijms231911023

**Published:** 2022-09-20

**Authors:** Giulia D. S. Ferretti, Julia Quarti, Gileno dos Santos, Luciana P. Rangel, Jerson L. Silva

**Affiliations:** 1Institute of Medical Biochemistry Leopoldo de Meis, National Institute of Science and Technology for Structural Biology and Bioimaging, National Center of Nuclear Magnetic Resonance Jiri Jonas, Universidade Federal do Rio de Janeiro, Rio de Janeiro 21941-901, Brazil; 2Department of Pharmaceutical Sciences, Universidade Federal Rural do Rio de Janeiro, Seropédica 23890-000, Brazil; 3Faculty of Pharmacy, Universidade Federal do Rio de Janeiro, Rio de Janeiro 21941-902, Brazil

**Keywords:** p53, protein aggregation, cancer

## Abstract

p53 is a tumor suppressor protein that is mutated in more than 50% of cancer cases. When mutated, it frequently results in p53 oncogenic gain of function (GOF), resulting in a greater tendency to aggregate in the phase separation and phase transition pathway. GOFs related to p53 aggregation include chemoresistance, which makes therapy even more difficult. The therapies available for the treatment of cancer are still quite limited, so the study of new molecules and therapeutic targets focusing on p53 aggregates is a promising strategy against cancer. In this review, we classify anticancer molecules with antiaggregation properties into four categories: thiol alkylating agents, designed peptides, agents with chaperone-based mechanisms that inhibit p53 aggregation, and miscellaneous compounds with anti-protein aggregation properties that have been studied in neurodegenerative diseases. Furthermore, we highlight autophagy as a possible degradation pathway for aggregated p53. Here, considering cancer as a protein aggregation disease, we review strategies that have been used to disrupt p53 aggregates, leading to cancer regression.

## 1. Introduction

The tumor suppressor protein p53, encoded by the *TP53* gene, is often referred to as a “guardian of the human genome”. It is responsible for the maintenance of DNA integrity, the inhibition of cell proliferation, and the prevention of tumorigenesis, promoting a coordinated response to cellular insults [1,2,3]. Given the enormous frequency of p53 mutations, which occur in more than 50% of all human cancer cases, it is no surprise that the modulation of mutant p53 (mutp53) is of great interest in cancer treatment.

Most of the mutations in the *TP53* gene are missense mutations, with hotspot mutation sites (R273, R248, R175, G245, G249, and R282), located in the p53 DNA-binding domain (DBD) [4]. These mutations can be categorized into contact mutations (which directly affect the interaction of p53 with DNA, but preserve protein stability and folding states in levels comparable to wild-type p53 (wtp53), represented by mutants R248Q and R273H) and conformational mutations, which cause changes in p53 structural stability at different levels, including the mutants R175H and R282W, which are largely destabilized and unfolded, and others such as G245S and R249S, which have local changes near the specific mutations [5,6,7]. Mutations in *TP53* lead to a loss of p53 canonical functions (loss of function, LOF) [8]. In addition, mutp53 exerts a dominant-negative (DN) effect by recruiting wild-type p53 (wtp53) and inhibiting its activity when expressed in the same cell [9]. The most impressive group of effects caused by p53 mutations are the gains of oncogenic function (GOFs) [10,11], in which mutp53 acts as an oncogene and promotes changes in the tumor interactome and transcriptome, leading to different effects related to cancer progression and depending on the mutation, different GOFs can be observed [10]. The effects produced include chemoresistance, increased tumor invasiveness, and metabolic reprogramming, among others [11,12].

The aggregation and deposition of misfolded proteins are involved in almost all neurodegenerative maladies, such as Alzheimer’s, Parkinson’s, and Huntington’s diseases [13]. The amyloid aggregation of p53 was only reported in 2003 [14], and prior to this, high p53 levels in vitro or in vivo were described as a consequence of accumulation, cytoplasmic retention, or denaturation [15,16,17]. Aggregation of wtp53 was first observed in in vitro and in vitro tumor cells [14,18,19,20]. In the last decade, p53 aggregates were identified in tumor samples from cancer patients and in several cancer cell lines, which established a correlation between mutp53 and p53 aggregation [18,21,22,23]. Both contact and conformational p53 mutants can form aggregates [24]. However, some mutants are more prone to aggregation than others; for example, the R248Q, R248W, R175H, and Y220C mutants have been found to aggregate in several tumor samples, whereas the ability of R273H and R249S mutants to form aggregates is debated [25,26]. Recently, it was shown that p53 aggregation is preceded by liquid–liquid phase separation [27,28]. These observations demonstrate the complexity of the process of p53 aggregation, which still needs further studies.

The interaction and coaggregation of aggregated mutp53 with other tumor suppressors, such as p63 and p73, [22], and chaperones [29], lead to GOF effects, resulting in the activation of cellular machinery for tumor initiation and/or progression [22,29,30,31]. Finally, the ability of mutp53 to form aggregates with prion-like properties has been described in vitro [18,32,33], and might be related to cancer progression [24]. Taken together, the influences exerted by aggregated mutp53 on the tumor cell and its surroundings, through its prion-like effect, appear to be an interesting target to be explored in the development of new anticancer drugs. The molecules studied to date and the underlying mechanisms are described in this review.

Targeted cancer therapies have gained increasing attention in recent years. The main goal of this type of therapy is to identify targets and effective drugs that bind specifically, increasing success and reducing undesirable effects [34]. Strategies aiming at mutp53 have three main goals: restoring wtp53 functions or/and degrading mutp53 [35,36,37], and targeting downstream mediators of mutp53 GOFs [38]. These goals are obtained through the use of a few classes of molecules: cysteine-binding compounds, Zn^2+^ chelators, and peptides [37]. In this review, we begin by discussing features of mutp53 and aggregation related to chemoresistance, a major drawback in cancer treatment. Subsequently, we describe the strategies specifically targeting mutp53 aggregation, clustered into four main groups, as described in Figure 1.

Finally, we discuss a promising strategy, still underexplored, to eliminate the deleterious effects of aggregated mutp53 through the promotion of its degradation by the autophagy of protein aggregates, named aggrephagy [39]. Since mutp53 presents different GOFs that promote cancer progression, the elimination or reduction of aggregated mutp53 levels seems to promote a reversion of the tumor cells to a less aggressive phenotype [40]. Autophagy is considered an important mechanism capable of eliminating unnecessary/dysfunctional organelles or proteins [41]. It is well known that autophagic pathways control mutp53 stability by regulating protein cellular levels and degradation [42]. However, aggrephagy in p53-positive cancer cells still needs to be further studied.

## 2. Chemoresistance and p53 Aggregation 

Resistance to anticancer drugs is considered the main obstacle to efficacious chemotherapy treatment, leading to poor prognosis of patients [21,31,40]. As discussed above, p53 mutations result in GOFs, contributing to cancer development and progression. These include a high-proliferation and low-apoptosis phenotype, increased cell migration, invasion and angiogenesis, metabolic reprogramming, and induced drug resistance [11].

The roles of mutp53 in the development of chemoresistance to cytotoxic agents involve: (1) increased activity of drug efflux transporters from the ATP-binding cassette (ABC) superfamily, such as P-gp [43,44,45], that actively expel multiple chemotherapeutic drugs across plasma membranes, decreasing their intracellular concentration; (2) increased apoptosis inhibition, reducing the cell death of cancer cells [46,47,48]; (3) improved DNA repair capacity, reducing the effect of therapies that cause DNA damage [49]; and (4) increased drug metabolism, inactivating the anticancer drugs [50]. However, the aggregated p53-based molecular mechanisms of chemoresistance are not fully understood, as evidenced in recent studies on resistance to cisplatin [51], carboplatin [40], and temozolomide [31], described below. 

Platinum drug resistance associated with p53 aggregation was identified in lung cancer cells (H1299) transfected with mutp53 (R282W) [51]. p53 aggregates formed by mutp53 (R282W) promote chemoresistance to cisplatin in lung cancer cells by activating the unfolded protein response (UPR) and upregulating endoplasmic reticulum protein 29 (ERp29) [51], which can be a marker of endoplasmic reticulum stress, impairing protein folding, trafficking, and secretion [52]. mutp53 (R282W) was shown to specifically bind to the promoter region of the *ERP29* gene, generating chemoresistance [51]. The authors described the *ERP29* gene as a possible target capable of sensitizing lung cancer cells bearing aggregation-prone mutp53 (R282W) to cisplatin treatment [51]. Interestingly, ERp29 expression, previously cited as a marker of chemoresistance caused by aggregation, can be suppressed by ReACp53, which resensitizes cells to cisplatin treatment [51].

Nevertheless, in the platinum drug field, carboplatin is also related to p53 aggregation in ovarian cancer stem cells (OCSCs) expressing wtp53 [40]. Mutp53 leads to LOFs, in addition to DN effects and GOFs, as previously mentioned. However, the expression of wtp53 in tumor cells may not always represent a favorable prognosis, such as in the case of p53 inactivation by excess MDM2 [53]. Additionally, wtp53 is as capable as mutp53 of forming aggregates, affecting protein functions and acting as a key factor for tumorigenesis [26]. Yang-Hartwich et al. (2015) [40], showed that both in vitro and in vivo OCSC experimental models expressing wtp53 demonstrated a curious mechanism in which p14ARF, a p53-positive regulator, inhibited MDM2-mediated p53 degradation, triggering an imbalance of p53 turnover, wtp53 aggregation, and resistance to carboplatin [40]. In a 3D model of MCF-7 breast cancer cells, wtp53 forms aggregates, and 5-fluorouracil (5-FU) treatment further increases wtp53 aggregation, partially inhibiting apoptosis and chemoresistance [54].

Resistance to temozolomide (TMZ) can be induced in glioblastoma, although the mechanism is not fully understood [55]. TMZ treatment induces DNA damage in human tumors; however, O6-methylguanine DNA-methyltransferase (MGMT) repairs DNA damage, contributing to drug resistance [52]. In an experimental model of glioblastoma, the mutp53 cell lines T98G (M237I) and U138-MG (R273H) both overexpress MGMT and are more resistant to TMZ treatment than the wtp53-containing cell line U87 [52]. Recently, it was reported that this link between mutp53 and chemoresistance is also correlated with amyloid formation in T98G cells expressing mutp53 (M237I) [31].

The studies reinforce the importance of finding new therapeutic approaches that could overwhelm chemoresistance effects exerted by mutp53 aggregation in cancer.

## 3. Molecules Used to Target p53 Aggregation

Several studies have explored p53 mutants as targets; however, therapeutic strategies directly or indirectly targeting mutp53 aggregates are still rarely found in the literature [56,57]. Since p53 mutations are present in more than 50% of all cancer cases and many p53 mutants are prone to aggregation in different types of cancer [58], p53 aggregation is a target with substantial therapeutic relevance [59]. On the other hand, p53 aggregation is also a complex event, considering that each mutant behaves differently, with different GOFs leading to tumor progression [59]. The search for molecules to break up mutp53 aggregates, leading to tumor destruction, is unveiling new prospects in anticancer therapy, enabling more efficient treatment and preventing chemoresistance. Compounds that target mutp53 aggregation are discussed below and are listed in Table 1.

### 3.1. Thiol Alkylating Agents

Alkylating agents can interact with thiol groups in mutp53 and refold the protein into a wild-type-like conformation [69,70]. The wtp53 DNA-binding domain (DBD) has 10 cysteine residues that are not equally reactive, and some of them are crucial for the correct folding and function of the protein [71]. When p53 is mutated, cysteine residues are exposed, resulting in protein misfolding [72,73]. This increases susceptibility to oxidation, which leads to the formation of intermolecular and intramolecular disulfide bridges, resulting in a variety of oligomeric forms, including monomers, dimers, and large aggregates [72,73]. Thiol alkylating compounds can bind to these reactive cysteines in the DBD through a nucleophilic reaction called Michael addition, inhibiting p53 aggregation. Thus, from a therapeutic perspective, the thiol groups of p53 cysteine residues are an important target for refolding mutp53 [70,74]. Two of these compounds, PK11000 and PRIMA-1, have been reported to prevent or disrupt p53 aggregation [33,60].

PK11000 is a small 2-sulfonylpyrimidine molecule that stabilizes the DBDs of both wtp53 and mutp53 proteins by covalent cysteine modification [60]. This compound is a mild and selective alkylating agent and specifically targets two cysteine residues, 182 and 277 [60]. Light scattering was used to demonstrate that PK11000 prevents p53 aggregation in a dose-dependent manner [60].

PRIMA-1, the acronym form of p53 reactivation with induction of massive apoptosis-1 (2,2-bis(hydroxymethyl)-1-azabicyclo[2,2,2]octan-3-one), was selected from a bank of 2000 low-molecular-weight compounds compiled by the National Cancer Institute (NCI) as a compound with the ability to target mutp53, promoting its reactivation [75,76]. A total of 14 missense p53 mutants were tested, and PRIMA-1 was able to restore 13 of them to a wild-type-like conformation; the only exception was the phenylalanine 176 mutation [75]. The large group of mutants affected by this compound includes both contact and structural mutants [75]. To increase the capacity of PRIMA-1 to induce apoptosis and to improve its membrane permeability, PRIMA-1 was methylated, generating PRIMA-1 MET (APR-246) [77]. Both PRIMA-1 and APR-246 have been shown to decrease proliferation and trigger apoptosis in vitro and in vivo in different types of cancer [33,75,76,78,79,80,81,82,83,84,85,86].

PRIMA-1 and APR-246 are prodrugs, and their common active metabolite is methylene quinuclidinone (MQ) [87]. MQ is a Michael acceptor that reacts with alkylating nucleophilic thiol groups and covalently binds to the nucleophilic –SH of p53 cysteine residues [87,88]. Cysteine 277 alone or together with cysteine 124 has been shown to be key for the stabilization and functional restoration of wt and mutp53 [89]. Techniques such as molecular modeling, high-resolution crystal structure analysis, and Nanomate Linear Ion Trap Orbitrap hybrid mass spectrometry (LTQ-MS) have revealed that the interaction may depend on the mutant, accessibility of the cysteine residue, and presence of DNA [87,89,90,91].

To uncover the mechanism by which PRIMA-1 reactivates p53, our group demonstrated that this compound is able to inhibit p53 aggregation [33]. DBD-based aggregation of both wt and mutp53 (R248Q) was prevented by PRIMA-1 and MQ in a dose-dependent manner, as shown by light scattering and thioflavin T (ThT) binding. Using the breast cancer cell line MDA-MB-231, which carries the R280K mutation, and the ovarian cancer cell line OVCAR-3, which carries the R248Q mutation, we also demonstrated the capacity of PRIMA-1 and MQ to disrupt mutp53 aggregation using immunofluorescence, immunoprecipitation, and size-exclusion chromatography. PRIMA-1 and MQ were also able to inhibit mutp53 prion-like properties in vitro [33].

To date, the molecular mechanisms of PRIMA-1 include the inhibition of aggregation with the reactivation of mutp53. Although APR-246 has not yet been reported to act in p53 aggregation, both APR-246 and PRIMA-1 are prodrugs that are converted into MQ under physiological conditions [77]. Thus, their mechanisms of action are expected to be the same.

Clinical trials using APR-246 alone or in combination with other drugs due to the ability to target mutp53 are underway (ClinicalTrials.gov Identifiers: NCT04383938, NCT04990778, NCT04419389, NCT03268382, NCT00900614, NCT02098343, NCT04214860, NCT02999893, NCT03072043, NCT03588078, NCT03745716, NCT03391050, NCT03931291) or have been concluded [92,93,94]. However, these studies did not address p53 aggregation.

### 3.2. Designed Peptides

The identification of p53 regions with a greater propensity to aggregate, forming amyloid structures, led to the design of new compounds that bind in a complementary way to these regions, restoring a wild-type-like conformation and disrupting mutp53 aggregation [22,58,75,95,96]. Software such as ZipperDB, TANGO, AGGRESCAN, and PASTA was used to predict regions that lead to p53 β-aggregation, with residues 250–257 (PILTIITL) found to be the region of p53 likely responsible for the formation of aggregates [96].

ZipperDB is a software program that attempts to find segments containing complementary sequences that form a steric zipper—two self-complementary beta-sheets [97]. ReACp53 is a peptide inhibitor specifically designed to target the p53 aggregation region determined by ZipperBD, and its ability to specifically inhibit the aggregation resulting from interactions with residues 252–258 (in the DBD) has been tested in vitro [25]. ReACp53 was modified with the addition of an N-terminal polyarginine cell-penetrating tag and a three-residue linker derived from the p53 sequence (RPI) [25]. In that study, cells were isolated from seven patients diagnosed with high-grade serous ovarian carcinomas (HGSOCs) bearing the P72R/R248Q, P72R/R196*h/R248Q, R248Q, P72R/I25S, V72M, P72R/Y234C, and P72R/Y326L mutations [25]. Using patient 1 samples (P72R/R248Q), the authors demonstrated that treatment with ReACp53 reduced the binding of PAb240 (an antibody to recognize unfolded p53) in a dose-dependent manner [25]. Using HGSOC organoids formed from either patient cells or previously established cell lines, the authors also showed that cells bearing mutp53 were more sensitive to ReACp53 treatment than cells expressing wtp53 (MCF-7 breast cancer cells) or null p53 (SKOV-3 ovarian cancer cells) [25]. Finally, in a xenograft model using the OVCAR-3 cell line (R248Q mutant) and MCF-7 cells (wtp53), treatment with ReACp53 decreased the size of xenograft tumors formed from mutp53 R248Q cells (OVCAR-3), which are known to be associated with p53 aggregation [25].

A study using prostate cancer cells with the P223L and V274F mutations (DU145 prostate cancer cells) and cells with the R273H and Q331R mutations (CWRR1 cells, derived from CWR22 castration-resistant prostate cancer cells) demonstrated through immunoprecipitation with Pab240 followed by immunoblotting with anti-p53 that there was less p53 in the unfolded fraction after treating native cell lysates with ReACp53 [63]. In vivo, using CWRR1 cell line-derived xenografts, an immunohistochemistry assay also showed less Pab240 binding after treatment with ReACp53 [63]. The CWRR1 cell line-derived xenograft immunohistochemistry assay also showed less Pab240 binding after treatment with ReACp53. Since there is no effective treatment for castration-resistant prostate cancer, these findings presented a new potential strategy to fight against this type of cancer

### 3.3. Chaperone-Based Mechanism for Preventing p53 Aggregation

Molecular chaperones have been intensively studied due to their importance for protein quality control in the cell [98,99]. Here, we highlight metallochaperones for their ability to maintain the correct folding of p53 [100], by binding metal ions such as zinc (Zn^2+^) to promote its delivery to p53 through protein–protein interactions [101].

The p53 DBD has one of the highest known zinc-binding affinities among eukaryotic proteins [100]. The tetrahedrally coordinated zinc ion binds to residues Cys176, His179 (L2), Cys238, and Cys242 (L3), which are present in two large loops that make up the p53 DBD [102]. Under physiological conditions, p53 acts as an important transcription factor, and binding to DNA is enhanced through the presence of zinc in the DBD (Holo-p53) [17]. Removal of the zinc ion (Apo-p53) promotes structural instability [103], hinders tetramer formation, and leads to protein inactivation [104], and widespread changes in the NMR spectrum of the protein with a reduction of ~3 Kcal·mol^−1^ in the apparent folding free energy are observed [105]. Additionally, overexpression of Zn^2+^-chelating protein (metallothionein) or Zn^2+^-chelating small molecules appears to modulate p53 folding and function [106].

Loss or changes in Zn^2+^ binding affinity may occur due to mutations that affect the tertiary structure of the protein, such as R175H [107]. Loss of Zn^2+^ exposes the amyloidogenic region of the protein (residues 251–257) or hydrophobic cavities, resulting in aggregation, as occurs in Y220C mutp53 [108]. Interestingly, it has also been reported that the structure of zinc-deficient mutp53 can be restored to the wild-type structure by re-establishing binding with Zn^2+^ [100]. Synthetic metallochaperones have been designed to restore p53 folding, but their effects on p53 aggregation have not been tested [109]. The mechanism is based on the transport of Zn^2+^ from the extracellular environment through the plasma membrane into cells, increasing intracellular Zn^2+^ levels to remetallate mutp53, and works for multiple mutants in cell culture and mouse models of cancer [107,109,110].

Two compounds with a similar structure to ThT, differing by an iodine atom, namely, L^I^ and L^H^, were compared as metallochaperones [64]. L^I^ was more efficient in inhibiting p53 aggregation and restoring its function through apoptosis induction with less toxicity to nontumoral cell spheroids. According to light scattering measurements, L^I^ prevented the aggregation of mutp53 (Y220C, in the DBD region), as confirmed by transmission electron microscopy. The gastric cancer cell line NUGC3, which expresses Y200C, showed weaker staining and merging between antiamyloid oligomer antibody (A11) and p53 (DO-1) signals after treatment with L^I^ in the immunofluorescence experiments. Additionally, the treatment was observed to increase Zn^2+^ uptake [64].

Molecular chaperones have also been studied due to their potential to target p53 aggregation [98,99]. Under normal conditions, when cells are exposed to stress, protein intermediates are sequestered to correct their folding and prevent aggregation [29,111,112]. Recently, chemical chaperones have been designed to restore structural mutp53 function [90,107,109,110,113]. Y220C mutp53 has a druggable crevice on its surface, distant from the DNA-binding region of the protein, which indicates that any inhibitory effect is not related to this region [7]. In silico methods were used to develop hit compounds to bind to the Y220C pocket, leading to protein stabilization and decreasing its aggregation rate [114,115,116,117,118,119]. Therefore, nine carbazole derivatives (with structures that bind and stabilize Y220C, functioning as molecular chaperones) were developed to target a subpocket of the crevice. Of these, only PK9318 was able to prevent mutY220C DBD- aggregation in a dose-dependent manner, as revealed by an in vitro light-scattering assay [65].

The possibility of using cellular chaperones to dissociate aggregates of p53 has also been explored. In a recent study, Huang and coworkers demonstrated that DAXX, a polyD/E protein, prevents and even reverses the aggregation of wtp53 and mutp53, as well as its main antagonist, MDM2 [120]. The restoration of the native conformations of aggregation-prone p53 mutants by DAXX suggests its potential as a cancer treatment [120].

### 3.4. Miscellaneous Compounds with Antiaggregation Properties Studied in Neurodegenerative Diseases

Proteinopathies described as “neurodegenerative diseases” are related to specific proteins with a high tendency to form amyloid structures, which is considered a hallmark of these pathological conditions [121,122,123]. Neuronal proteins causing protein misfolding diseases include tau and β-amyloid, both related to Alzheimer’s disease; α-synuclein, related to Parkinson’s disease; and huntingtin, related to Huntington’s disease [124]. When aggregated, these proteins show a toxic effect on neuronal cells, leading to cell damage or death [124]. Recombinant p53 aggregates have also shown toxicity to some degree (Ano Bom, 2012), but their main role appears to be to contribute to cancer cell survival and tumor maintenance through p53 GOFs [22]. Molecules that disassemble the protein aggregates involved in neurodegenerative diseases have been tested for their ability to reduce p53 aggregation due to their amyloid properties [61,66,67,68]. As mentioned before, p53 aggregates exhibit prion-like behavior, and this characteristic may also be correlated with neurodegenerative diseases [24]. Thus, a molecule with anti-aggregation properties could be a good candidate to treat both types of diseases. We describe below the literature available on antiamyloid aggregation compounds tested on p53.

This category includes the small stress molecules arginine analogs [68], and acetylcholine chloride [66]. In vitro experiments using ThT binding and dynamic light scattering techniques showed their ability to prevent p53 aggregation [66,68]. Arginine analogs prevent the aggregation of the QRPILTIITL peptide (p53 248–257), a mimetic peptide of the R248Q mutant [68], and acetylcholine chloride prevents the aggregation of the same sequence of p53 with another mutation, R248W (WRPILTIITL) [66]. In a breast cancer cell model bearing full-length R175H mutp53 (SK-BR-3 cells), polyarginine showed the capacity to restore wtp53 functions, according to its ability to increase p21 levels, as detected by immunofluorescence [68]. Although the increase in p21 expression after treatment is indicative of p53 reactivation, further experiments should be performed to strengthen the correlation between increased p21 expression after treatment and the disruption of intracellular aggregates.

Resveratrol (trans-3,4′,5-trihydroxystilbene), which has been reported to prevent aggregation and inhibit aggregate toxicity of various amyloid proteins, including transthyretin (TTR), islet amyloid polypeptide (IAPP), α-synuclein and amyloid-β (Aβ-42) peptide [125,126,127,128,129], was shown to inhibit p53 aggregation in a cancer model [67]. Da Costa et al. (2018) showed that resveratrol prevents aggregation of the DBD region using mutp53 (R248Q) and wtp53 in a dose-dependent manner through light scattering. For other resveratrol derivatives, pterostilbene and piceatannol, much higher concentrations were necessary [67]. Furthermore, resveratrol was tested in MDA-MB-231 (R280K) and HCC-70 (R248Q) mutp53 breast cancer cell lines, which were more sensitive to resveratrol treatment than the wtp53 cell line MCF-7 [67]. Resveratrol also reduced colocalization between p53 (DO-1) and amyloid oligomers (A11) in immunofluorescence experiments in mutp53 cell lines [67]. Lower levels of p53 aggregates were also detected in histological sections of BalbC/Nude mice xenografted with MDA-MB-231 tumors and treated with resveratrol [67].

Oligopyridylamide-based α-helix mimetics were shown to inhibit the aggregation of Aβ peptide, which is involved in Alzheimer’s disease, and IAPP, which is involved in type II diabetes [130,131,132,133]. Ten oligopyridylamides (ADH 1–10) were tested in a p53 aggregation model [61]. According to the ability to bind to ThT and the results of transmission electron microscopy, ADH-6, which has cationic properties, was shown to be the most efficient at inhibiting amyloid aggregation of a p53 peptide (composed of residues 248 to 273, which includes the p53 DBD aggregation subdomain-residues 251–258) with the R248W mutation [61]. Using cells with the same R248W mutation in p53 (MIA PaCa-2 pancreatic cancer cells), these authors also showed the ability of ADH-6 to disaggregate full-length endogenous p53 [61]. Thioflavin S (ThS) was used, and confocal fluorescence microscopy showed that after treatment with ADH-6, ThS and Pab240 colocalization almost completely disappeared, which reinforces the idea that ADH-6 can disaggregate p53 amyloids [61]. Experiments in mice xenografted with MIA PaCa-2 cells also showed less Pab240 labeling through immunohistochemistry [61].

Heterobifunctional molecules such as proteolysis-targeting chimera (PROTAC), autophagosome-tethering compound (ATTEC), and autophagy-targeting chimera (AUTOTAC), have emerged as new approaches to target protein degradation in both in vivo and in vivo models of neurodegenerative diseases [134,135]. They are composed of three chemical regions: two ligands joined by a linker, where one domain targets the protein of interest (POI), while the other one directs the protein to specific degradation pathways [134,135]. Mutp53 aggregates have not yet been targeted as a POI using this chemical degradation tool. However, the promising results in neurodegenerative diseases can unveil a new path for designing new molecules targeting mutant p53 aggregates, preferably with delivery to autophagy degradation, since these protein aggregates cannot be directly degraded via proteasome [39].

## 4. Relationship between mutp53 and Autophagy

Targeting mutp53 protein for degradation, rather than reactivating it, might be another strategy for drug development since the GOFs promoted by mutp53 are responsible for a plethora of effects that promote cancer progression [136,137]. Beyond stimulation of mutp53 proteolysis through the ubiquitin–proteasome system, inducing degradation of mutp53 by autophagy is also a useful strategy [42]. Mutp53 is, in most cases, highly expressed and stabilized in tumor cells. These proteins escape ubiquitination and degradation due to upregulation of the HSP90 chaperone machinery which, in turn, inhibits the activity of the E3 ubiquitin ligases MDM2 and CHIP [138,139]. Two different types of autophagic degradation have been described for mutp53 proteins: chaperone-mediated autophagy and macroautophagy [140]. Chaperone-mediated autophagy is a selective process with specific machinery components wherein the chaperone HSC70/HSPA8 binds protein targets containing the amino acid sequence “Lys-Phe-Glu-Arg-Gln” (called the KFERQ sequence) and transports them to the lysosome. In macroautophagy, generally called “autophagy” (and hereafter referred to as autophagy), autophagosomes (double-membrane vesicles) act as intermediates that transport cytoplasmic cargo into lysosomes [141]. During the investigation of autophagic activity, autophagosome accumulation is measured, for example, by quantifying LC3-II (which is anchored to the autophagosome membrane) [142]. Then, it is necessary to determine autophagic flux by measuring autophagy receptors, which promote autophagosome formation around specific substrates [142]. The most studied autophagy receptor is p62. At the end of the autophagic process, p62 is degraded along with other cargo proteins. If autophagic flux is enhanced (evidenced by increased p62 degradation), autophagy-inducing signals, such as mTORC1 activity, should be assessed [142]. mTOR is the most important negative regulator of autophagy, as it represents a point of convergence between complex signaling pathways controlling autophagy [143].

The relationship between p53 and autophagy has been shown to be extremely complex: while the autophagic pathway is regulated by p53 signaling (in general, wtp53 acts as a pro-autophagic factor, but mutp53 displays a suppressive role in autophagy); furthermore, p53 is also a target protein of autophagy [140]. Although there is little evidence that autophagy plays a role in wtp53 degradation [144], the autophagy pathway is important for controlling the stability of mutp53 under different types of stress [42]. In most cases, the therapeutic goal for tumors with mutp53 is autophagy induction to promote the degradation of mutp53 [42].

### Degradation of p53 Aggregates by Autophagy, Known as Aggrephagy

Mutp53 degradation via autophagy is a widely discussed topic in the scientific literature [140]; however, the elimination of misfolded or aggregated p53 by autophagy is still not extensively explored, despite its promise. The removal of p53 aggregates could sensitize cancer cells, even without restoring p53 function [10].

Aggregation-prone mutp53 mediates the sequestration of various tumor suppressors, including wtp53, p63, and p73 [22], in intracellular aggregates that are too large to fit in the gated channels of the proteasome, preventing their degradation by the proteasome [145]. These large cytoplasmic inclusion bodies, which act as disposal devices that will eventually be degraded via autophagy, are called aggresomes [146]. The presence of mutp53 in these cytosolic structures is important evidence that autophagy is a key route for mutp53 aggresome degradation. Specific autophagy of aggresomes or misfolded proteins is known as aggrephagy [39] (Figure 2).

In aggrephagy, autophagy receptors associate with polyubiquitylated misfolded proteins (as well as their aggresomes) and LC3-II, facilitating selective cargo recognition and targeting these protein aggregates for degradation by autophagy [147]. In addition to p62, other aggrephagy receptors, such as NBR1, OPTN, TOLLIP, and CCT2, have been described in neurodegenerative diseases [148,149]. Furthermore, they are usually not exclusively involved in aggrephagy and are also involved in other types of selective autophagy, such as mitophagy (which removes aged and damaged mitochondria) [150]. Among these receptors, the most specific receptor for aggrephagy seems to be the chaperonin subunit CCT2; however, it still needs to be tested in experimental cancer models [149]. Thus, the molecular mechanism of cargo selection during aggrephagy in p53-expressing cancer cells needs to be further elucidated.

Maan and Pati (2018) [151], explored the mechanism underlying the degradation of mutp53 aggregates in hypoxia. CHIP displays E3 ligase activity, degrading its substrate proteins, including mutp53, via proteasomal and lysosomal pathways [42]. Maan and Pati (2018) [151], also demonstrated that CHIP could differentially degrade aggregated mutp53 (R110P, R175H) through K63-linked polyubiquitination via autophagy, suggesting that CHIP has a tumor suppressor function in hypoxic conditions.

The evaluation of functional autophagy with defective aggrephagy is also a challenge [150]. Biel et al. (2020) [150] used an experimental model in which mitochondrial dysfunction (mitophagy) leads R280K mutp53 to form aggregates that accumulate in breast cancer cells (MDA-MB-231) due to insufficient degradation of aggregated protein (characterizing defects in aggrephagy). These authors showed that the autophagy receptors TAX1BP1 and NDP52 were degraded by autophagy (indicating functional autophagy) even when p53-positive protein aggregates accumulated (indicating aggrephagy inhibition), showing that these are better markers for monitoring autophagy than conventional methods using p62 staining [150]. Therefore, assessment of these autophagy biomarkers in patients who harbor mutp53 might contribute to personalized treatment options in the future.

Researchers have studied aggrephagy as a target in p53-expressing cancer cells in works using different approaches, such as the colocalization of protein aggregates with LC3-II or p62 protein as an aggrephagy marker [145]; the knockdown or inhibition of autophagy-related proteins to test whether p53 aggregates are no longer degraded [152]; and the coimmunoprecipitation of p53 protein aggregates with autophagy-related proteins to investigate whether the interaction between these proteins is reduced as autophagy increases, suggesting their degradation [153].

Rodriguez et al., 2012 [145] evaluated the effect of a dietetic intervention (carbohydrate-free diet) on K132Q mutp53 aggregate degradation. In that study, the in vitro experimental conditions were generated by cultivating the breast cancer cell line BT20 in regular medium or medium lacking glucose. The authors found that glucose restriction led to mutp53 colocalization with p62-positive aggregates, suggesting the occurrence of aggrephagy [145,154]. Aggarwal et al. (2016) [152] used another dietary approach and demonstrated that a food compound isolated from cruciferous vegetables (phenethyl isothiocyanate, PEITC) depletes R175H mutp53 aggregates in the breast cancer cell line SK-BR-3 through autophagy. Furthermore, PEITC also rescued mutp53 activity by restoring wtp53 tumor-suppressive function, leading to higher sensitivity to MDM2-dependent proteasome degradation [152].

The bioactive components of several plants used in traditional Chinese medicine also seem to be promising candidates for p53-targeting therapeutic strategies. In a recent study, emodin presented anticancer effects by removing p53 protein aggregates, formed independently of *TP53* mutations [153]. Therefore, aggrephagy studies should not be limited to experimental models with mutp53. p53 aggregation also occurs in wtp53 cancer cells because other cellular conditions (in addition to mutation status) may change p53 conformation, such as oxidative stress [26]. Haque et al. (2018) [153] also demonstrated that emodin eliminates wtp53 aggregates via autophagy in the lung cancer cell line A549. In turn, disaggregation of wtp53 probably restores its normal protein function, inducing wtp53 translocation into the nucleus, which further increases autophagy levels by inhibiting the mTOR pathway [153].

## 5. Conclusions

The studies discussed in this review reveal promising strategies to target p53 aggregation to treat cancer (Figure 1). Still, several questions need to be answered for these approaches to reach the clinics. One of the greatest difficulties is the fact that each mutant behaves in a specific way and consequently results in a distinct aggregation pattern. In addition, each type of tumor has its own specific characteristics. We anticipate that molecules that target p53 aggregation can be used in association with other anticancer compounds.

In general, a few common steps to describe new mutp53 aggregation inhibitors are found in these papers, briefly summarized here: first, a new molecule must be proposed, then it must be tested in vitro to prove its capacity to prevent mutp53 aggregation by targeting a heterologous protein or region of the protein. The second step is to use a cellular model to test whether the new molecule can disrupt p53 aggregation in 2D and/or 3D cell models. If it can, the outcomes of p53 aggregation inhibition in the model studied are elucidated. Finally, to progress to clinical trials, it is necessary to determine whether these same mechanisms occur in in vivo models with the cells used in the second step.

Another challenge in understanding the mechanisms of the compounds in human cells and their in vivo outcomes is the type of model usually used, which include Balb c/Nude mice, with a limited immune system. New models which comprise the immune response after drug treatment should be developed for this case. In addition, new drug delivery systems are necessary to improve drug effects and accuracy, as was discussed by Kanapathipillai (2018) [59].

Finally, with this review, our hope is to provide an overview of the strategies to use p53 aggregation inhibition and degradation as promising targets for chemotherapy, so that in the coming years we can reach a large number of patients and improve their life expectancy and prognosis.

## Figures and Tables

**Figure 1 ijms-23-11023-f001:**
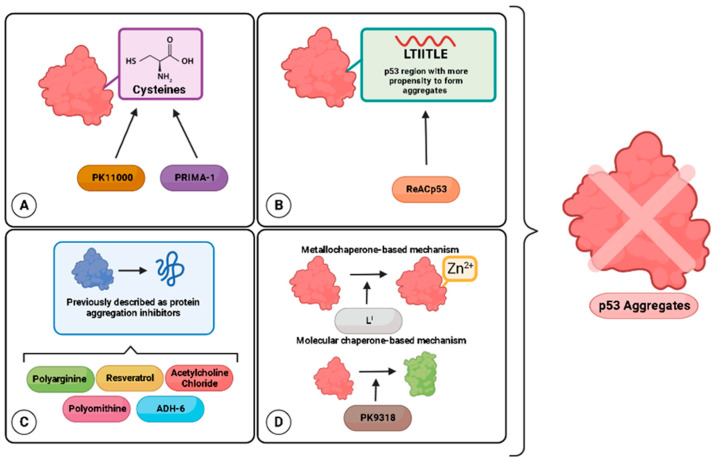
Strategies to target p53 aggregation as anticancer candidates. (**A**) Thiol alkylating agents targeting mutp53 cysteines. (**B**) Designed peptides that bind in a complementary way to regions of p53 with a greater propensity to aggregate. (**C**) Miscellaneous compounds with anti-protein aggregation properties previously tested in neurodegenerative diseases. (**D**) Compounds with metallochaperone- or molecular-chaperone-based mechanisms.

**Figure 2 ijms-23-11023-f002:**
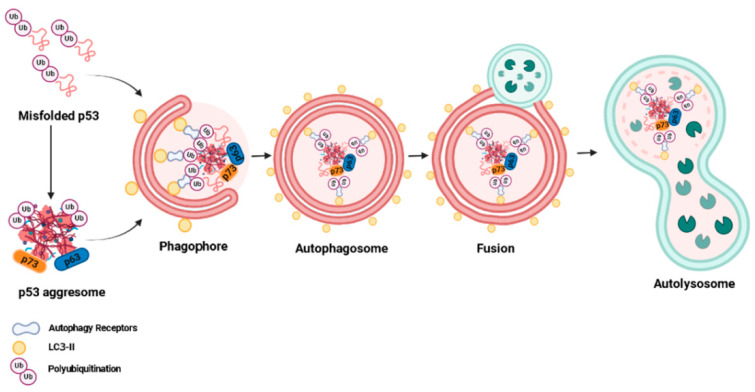
Aggrephagy mechanism described in p53-expressing cancer cells. Misfolded p53 and its aggresomes are recognized by autophagy receptors, such as p62, and are then directed to autophagic degradation. The autophagy process involves the formation and elongation of the phagophore membrane; engulfment of protein aggregates by the phagophore membrane to form autophagosomes; fusion of the autophagosome with lysosomes to form autophagolysosomes; and degradation of protein aggregates by proteolytic enzymes in lysosomes.

**Table 1 ijms-23-11023-t001:** Published strategies and compounds tested for mutp53 aggregation inhibition.

Structure/Compound	Mut/WT Aggregates	Model	Outcomes	Refs
Therapeutic strategy: thiol alkylating agents targeting mutp53 cysteines				
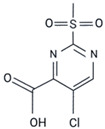 PK11000	Y220C	in vitro	- Prevents aggregation of Y220C mutp53c.	[60]
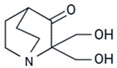 PRIMA-1	Wild-type, R248Q and R280K	in vitro	- Prevents aggregation of wtp53c and R248Q. - Inhibits seeding promoted by R280K in cell lysates. - Disrupts mutp53 aggregation in cells. - Induces apoptosis.	[33]
Therapeutic strategy: designed peptides for complementary binding to regions of p53 with a greater propensity to aggregate.				
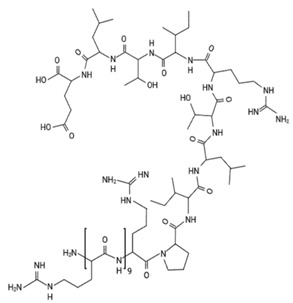 ReACp53	R248Q, P223L, V274F, R233H, Q331R, and H1299 cells transfected with R282W and R248W	in vitro and in vivo	- Prevents aggregation of the peptide LTIITLE (p53 residues 252–258). - Less PAb 240 staining; - Shows significant SDS resistance. - Significantly increases p21, GADD45B, PUMA, NOXA and DRAM1 mRNA. - Causes tumor regression and reduction of Ki67-positive cells. - Association with carboplatin shows increased mouse survival and induction of apoptosis.	[25,61,62,63]
Therapeutic strategy: compounds with metallochaperone- or molecular chaperone-based mechanisms				
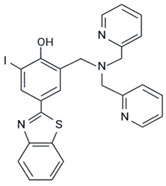 L^I^	Y220C	in vitro	- Prevents aggregation of Y220C mutp53c. - Reduces the colocalization of p53 and amyloid oligomers. - Increases the Zn^2+^ influx, indicating Zn-metallochaperone activity	[64]
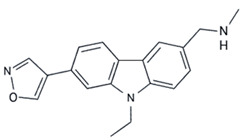 PK9318	Y220C	in vitro	- Prevents aggregation of Y220C mutp53c; - Reduces cell viability.	[65]
Therapeutic strategy: miscellaneous compounds with anti-protein aggregation properties previously tested in neurodegenerative diseases.				
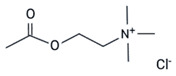 Acetylcholine Chloride	R248W	in vitro	- Prevents aggregation of the peptide WRPILTIITL bearing the R248W mutant.	[66]
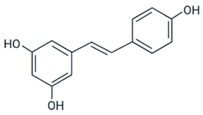 Resveratrol	wtp53 and R248Q	in vitro and in vivo	- Prevents aggregation of wtp53c and R248Q mutp53c. - Disrupts mutp53 aggregation in cells. - Prevents cell migration and proliferation.	[67]
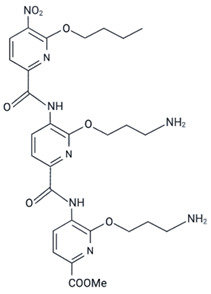 ADH-6	R248W	in vitro and in vivo	- Prevents aggregation of the mutant mimetic peptide R248W (WRPILTIITLEDSSGNLLGRNSFEVR). - Disrupts and reactivates mutp53 aggregates. - Induces apoptosis and changes cell cycle with more cells in the G0/G1 phase. - Shows tumor regression and less PAb 240 staining of tumor tissue.	[61]
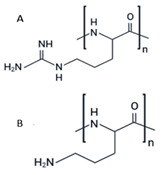 Polyarginine (A) and polyornithine (B)	R248Q	in vitro	- Prevents aggregation of the peptide QRPILTIITL bearing the R248Q mutant. - Inhibits proliferation in cells. - Has no toxicity to normal cells. Polyornithine increases p21 levels in cells.	[68]

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
