# Peer review of "Anticancer Therapeutic Strategies Targeting p53 Aggregation"

_ijms, 2022, doi:10.3390/ijms231911023_

Round 1

Reviewer 1 Report

In the present review, the authors describe the literature available on the mutant p53 aggregation resulting in a specific gain of function while losing normal function. The abnormal gain of function in mutant p53 is chemoresistance. Therefore, it would be ideal to study the molecules that can disrupt the aggregation. There are many limitations to this review article. Please see my concerns below.

 The major drawback is that this review describes many different angles without direct connections. Mostly jumping from one topic to another with a sudden shift without obvious connection.

 In several places, the references are missing. It starts with the first paragraph and first line of the review and follows throughout the review.

 In line number 53, it was written that mutant p53 aggregation results in the loss of function. Does it mean that mutant p53 without aggregation still contains the actual function?

 How the authors believe that heterobifunctional PROTACs to molecular glues are not superior to those described in lines 61-69 of the paragraph. Describe this in more detail while comparing it in the review.

 The authors should include all examples in which it was discussed and compared to a neurodegenerative disorder.

Line number 76 why jump to Autophagy? There are many more processes in the cell that can be targeted.

In section 2, the reported drugs are the only drugs found to have resistance to p53-specific cancers.

Section 3.3 why directly jump to neurodegenerative disorders, does not make much sense.

 Report the specific proven and validated research articles that describe the mechanism of Figure 2. Is it the general mechanism or is it relative to p53 degradation?

Author Response

Reviewer 1

In the present review, the authors describe the literature available on the mutant p53 aggregation resulting in a specific gain of function while losing normal function. The abnormal gain of function in mutant p53 is chemoresistance. Therefore, it would be ideal to study the molecules that can disrupt the aggregation. There are many limitations to this review article. Please see my concerns below.

The major drawback is that this review describes many different angles without direct connections. Mostly jumping from one topic to another with a sudden shift without obvious connection.

Answer: We thank the reviewer for the comment. We have made substantial changes in the text to improve the connection between the topics. In addition to rewriting, we have included new paragraphs to the topics.

In several places, the references are missing. It starts with the first paragraph and first line of the review and follows throughout the review.

Answer: We appreciate the observation. We have revised the references cited throughout the text.

In line number 53, it was written that mutant p53 aggregation results in the loss of function. Does it mean that mutant p53 without aggregation still contains the actual function?

Answer: We apologize, the idea was poorly described. The sentence was revised. Actually, it is p53 mutation that leads to its loss of function.

How the authors believe that heterobifunctional PROTACs to molecular glues are not superior to those described in lines 61-69 of the paragraph. Describe this in more detail while comparing it in the review.  The authors should include all examples in which it was discussed and compared to a neurodegenerative disorder.

Answer: The topic “Miscellaneous compounds with anti-aggregation properties studied in neurodegenerative diseases” was intended to present the compounds used before in neurodegenerative diseases that were actually tested on p53 aggregation. We appreciate the reviewer’s comments and think that this could be an excellent strategy to degrade p53 aggregates. For this reason, we included a paragraph explaining these strategies briefly and proposing that aggregates must be better degraded by molecules that deliver them to autophagy, since these aggregates are preferentially addressed to that pathway.

Line number 76 why jump to Autophagy? There are many more processes in the cell that can be targeted.

Answer: We appreciate the reviewer’s comments. We have improved the connection of the text with this section. The other processes that could be modulated in cancer cells (such as DNA repair, cell cycle, and apoptosis) would not be directly related to p53 aggregates, unlike autophagy which could be a possible route for degradation of these aggregates. In our view, autophagy-driven mechanisms can be promising approaches to tame mutant p53 deleterious effects. We explain in section 4.1 that GOF effects can be contended by reducing mutant p53 levels through aggrephagy in the tumor cell.

In section 2, the reported drugs are the only drugs found to have resistance to p53-specific cancers.

Answer: We thank the reviewer for the opportunity to clarify this point. Actually, in this section, we highlighted aggregated p53-based molecular mechanisms of chemoresistance and not of p53 mutations in general. To make this clear in the text, we have added introductory sentences in which we mention other resistance mechanisms related to p53 mutations, in addition to those already related to p53 aggregation.

Section 3.3 why directly jump to neurodegenerative disorders, does not make much sense.

Answer: This topic is meant to gather information about the compounds with antiamyloid aggregation properties that were tested previously on neurodegenerative diseases, which are frequently used models to evaluate anti-aggregation compounds. We have added a few introductory sentences to improve the insertion of this topic in the review text.

Report the specific proven and validated research articles that describe the mechanism of Figure 2. Is it the general mechanism or is it relative to p53 degradation?

Answer: Although aggrephagy occurs in a similar way in different experimental models, in Figure 2, we chose to represent aggrephagy in p53-positive cancer cells. This is why we highlighted the p62 autophagy receptor, since this was the only one evaluated in this experimental model so far (Rodriguez et al., 2012 doi:10.4161/cc.22778; Choudhury et al., 2013 doi:10.4161/cc.24128). However, it should be noted that this receptor would not necessarily be the most specific or even the only one possible for these cells, as we discussed in section 4.1 of the paper. For this reason, to avoid confusing the reader, we chose to remove the word “p62” and replace it with a more general term such as “autophagy receptor”.

Reviewer 2 Report

The article by Ferretti et al., entitled “New anticancer therapies targeting p53 aggregation Giulia” claim to propose new therapeutic approaches to target p53 aggregation. And they have concluded these strategies as a promising avenue for developing personalized therapies against cancer.

 In reality they have summarized information on some anticancer molecules and put them in four categories to prevent p53 aggregation. It still needs some attention and could be useful for readers, however, most of the manuscript is incomplete or misleading. To begin with, these are not new therapies but some existing strategies, thus the title should be changed to something like “small molecule modulated prevention of p53 aggregation”.

Further, the Introduction and section 2 needs to be written with better citations. First the author should mention the similar existing therapies which target p53 for anticancer drug development. There are ‘SEVERAL’ on p53 in cancer. Then the relevance of chemoresistance and p53 aggregation should be established. They should give proper explanation why this review is different from the existing ones.

 Table one can be made more readable. Authors may keep the structure and briefly mention investigated pathway and ref. They have mentioned 4 strategies(categories), so they mention them and put the explanation in the text of the manuscript.

 Are any of these compounds in clinical trial? Or is there any study relevant to their review. Instead of abstractly mentioning that there are several questions that need to answered, put them as a section mentioning challenges.

 Finally, please do not confuse readers by mentioning terms like ‘personalized therapies against cancer’. It is a huge field in itself which is in infancy. However, it is not appropriate to include personalized cancer therapies without mentioning a single word about the field in the manuscript.

Author Response

The article by Ferretti et al., entitled “New anticancer therapies targeting p53 aggregation Giulia” claim to propose new therapeutic approaches to target p53 aggregation. And they have concluded these strategies as a promising avenue for developing personalized therapies against cancer.

In reality they have summarized information on some anticancer molecules and put them in four categories to prevent p53 aggregation. It still needs some attention and could be useful for readers, however, most of the manuscript is incomplete or misleading. To begin with, these are not new therapies but some existing strategies, thus the title should be changed to something like “small molecule modulated prevention of p53 aggregation”.

Answer: We thank the reviewer for his opinion. In order to correct these mistakes, we tried to reorganize each topic during the review process, making considerable changes throughout the text. Thus, these modifications, including the title, are incorporated in the revised version of the paper in word format.

Further, the Introduction and section 2 needs to be written with better citations. First the author should mention the similar existing therapies which target p53 for anticancer drug development. There are ‘SEVERAL’ on p53 in cancer. Then the relevance of chemoresistance and p53 aggregation should be established. They should give proper explanation why this review is different from the existing ones.

Answer: We are grateful for these observations. We have tried to improve citations and to complement the text with the requested information.

Table one can be made more readable. Authors may keep the structure and briefly mention investigated pathway and ref. They have mentioned 4 strategies(categories), so they mention them and put the explanation in the text of the manuscript.

Answer: We thank the reviewer for the suggestion. We have reformulated Table 1, as recommended.

Are any of these compounds in clinical trial? Or is there any study relevant to their review.

Answer: We have included the citation and discussion on clinical trials with APR-246, but we chose not to include for the other compounds, such as resveratrol because they are not being tested with a focus on mutant p53 in the available clinical trials.

Instead of abstractly mentioning that there are several questions that need to answered, put them as a section mentioning challenges.

Answer: We appreciate the reviewer’s suggestion. We made changes in the text to highlight the challenges.

Finally, please do not confuse readers by mentioning terms like ‘personalized therapies against cancer’. It is a huge field in itself which is in infancy. However, it is not appropriate to include personalized cancer therapies without mentioning a single word about the field in the manuscript.

Answer: We apologize for the misuse of the expression. The word ‘personalized’ has been replaced by ‘targeted’.

Round 2

Reviewer 1 Report

No more queries

Reviewer 2 Report

Its much better version and can be accepted.